# Self-healable polymer complex with a giant ionic thermoelectric effect

Dong-Hu Kim [1], Zico Alaia Akbar[1], Yoga Trianzar Malik[2], Ju-Won Jeon [2] ✉ & Sung-Yeon Jang [1,3] ✉

In this study, we develop a stretchable/self-healable polymer, PEDOT:PAAMPSA:PA, with remarkably high ionic thermoelectric (iTE) properties: an ionic figure-of-merit of 12.3 at 70% relative humidity (RH). The iTE properties of PEDOT:PAAMPSA:PA are optimized by controlling the ion carrier concentration, ion diffusion coefficient, and Eastman entropy, and high stretchability and self-healing ability are achieved based on the dynamic interactions between the components. Moreover, the iTE properties are retained under repeated mechanical stress (30 cycles of self-healing and 50 cycles of stretching). An ionic thermoelectric capacitor (ITEC) device using PEDOT:PAAMPSA:PA achieves a maximum power output and energy density of 4.59 μW·m⁻² and 1.95 mJ·m⁻², respectively, at a load resistance of 10 KΩ, and a 9-pair ITEC module produces a voltage output of 0.37 V·K⁻¹ with a maximum power output of 0.21 μW·m⁻² and energy density of 0.35 mJ·m⁻² at 80% RH, demonstrating the potential for a self-powering source.

Thermoelectric (TE) devices have received significant attention for energy generation (or harvesting) from otherwise-wasted low-quality heat, owing to their unique advantages (no moving parts, no pollution, light weight, easy production, etc.) over traditional heat engines[1–3]. While conventional TE generators use electrons/holes as charge carriers for continuous power generation, the recently emerging ionic thermoelectric capacitor (ITEC) is operated based on ion diffusion in ionic TE (iTE) materials[4–6]. In iTE materials, the ions separated by a thermal gradient cannot be transferred through a circuit; thereby, the ITEC relies on charging/discharging via an external circuit, which is often observed in conventional capacitors[7–9]. iTE materials have advantages in terms of output voltage (~tens of mV·K⁻¹) over conventional electronic TE materials (~tens of μV·K⁻¹), which might be essential in self-powering wearable devices[10].

The performance of an iTE material is fundamentally dependent on the Soret effect (or thermophoresis), which relies on the spontaneous transport of particles under the action of a thermal gradient[4,9,11]. The fourth law of thermodynamics (Onsager reciprocal relations) describes the Soret effect as heat flow leading to matter flow[12]. The performance of iTE materials is characterized by the dimensionless ionic figure-of-merit:

$$ZT_i = \frac{S_i^2 \sigma_i}{\kappa} T \tag{1}$$

where $S_i$ is the ionic Seebeck coefficient, representing the thermovoltage under a definite temperature difference, $\sigma_i$ is the ionic conductivity, $\kappa$ is the thermal conductivity, and $T$ is temperature[8]. To achieve high $ZT_i$, a simultaneous improvement in $S_i$ and $\sigma_i$ is crucial. According to Eastman, the difference in the structural entropy of the ion solvation shell at the hot and cold sides induces thermophoresis, which builds an electric field between the two sides[6,10,11,13]. The $S_i$ of an iTE material with one type of ion ($S_x$) can be expressed as,

$$S_x = S_i^* \cdot q^{-1} \tag{2}$$

where $S_i^*$ is the Eastman entropy of the ion, and $q$ is the electric charge of the ion. However, in the case of iTE materials with more than one

[1]School of Energy and Chemical Engineering, Ulsan National Institute of Science and Technology (UNIST), 50 UNIST-gil, Ulsan 44919, Republic of Korea. [2]Department of Chemistry, Kookmin University, 77 Jeongneung-ro, Seongbuk-gu, Seoul 136-702, Republic of Korea. [3]Graduate School of Carbon Neutrality, Ulsan National Institute of Science and Technology (UNIST), 50 UNIST-gil, Ulsan 44919, Republic of Korea. ✉e-mail: jwjeon@kookmin.ac.kr; syjang@unist.ac.kr

type of carrier, $S_i$ is determined as,

$$S_i = \frac{\Sigma q S_i^* n_i D_i}{\Sigma q^2 n_i D_i} \qquad (3)$$

where $n_i$ is the bulk ion carrier concentration, and $D_i$ is the ion diffusion coefficient[14,15]. This equation can be simplified as,

$$S_i = I_{i,net} \cdot R_{i,net} \qquad (4)$$

in which the net ionic thermocurrent ($I_{i,net}$) is destructive and the net ionic resistance ($R_{i,net}$) is constructive (Supplementary Note 1). Hence, to improve $S_i$, it is vital not only to accomplish a high $S_i^*$ but also to optimize $I_{i,net}$. Although the relationship between $S_i$ and the structural entropy is still not widely investigated, there are some clues from previous studies that deal with the thermodynamical interpretation of the ion-solvent interaction[16–23]. First, in the case of hydration, the structural entropy ($\Delta S_{struc}$) is determined by the hydration entropy of ions ($\Delta S_{hyd}^\infty$, Supplementary Note 2)[21]. For instance, the hydration enthalpy ($\Delta H_{hyd}$) is highly negative for most ions, whereas the sign of the hydration entropy ($\Delta S_{hyd}$) depends on the ions[21,22]. The sign of $\Delta S_{hyd}$ for the ions is typically distinguished in the terminology of kosmotropic and chaotropic ions, depending on how the structure of water molecules is altered by the ions[21–23]. A kosmotropic ion has a high charge density and thus tends to construct structured solvation shell and surroundings (structure-making ion), whereas a chaotropic ion has opposite characteristics (structure-breaking ion)[22]. The proton is a representative kosmotropic ion with a negative $\Delta S_{hyd}$ like other low-period cations (e.g., Li and Na ion)[21,24]. Moreover, the $\Delta S_{hyd}$ of protons decreases even more at higher temperatures[24], which induces the thermodiffusion of protons from the hot side to the cold side (positive $S_i$)[23]. Thus far, the reported iTE materials, based on various solid and quasi-solid electrolyte systems, have achieved $ZT_i$ of 0.75–6.1, and they are neither stretchable nor self-healable[6,16,25–32]. Further design of iTE materials with high $ZT_i$ will be crucial to realize wearable self-powering devices.

Moreover, complex mechanical properties are also demanded to practically apply the iTE materials in wearable electronics, such as biosensors, artificial skins, and smart clothes[10,33]. In addition to mechanical robustness (Young's modulus ($Y$) of 0.5–1 MPa), sufficient stretchability and spontaneous self-repairing ability are required[34–36]. In a previous study, Fang et al. demonstrated a polyurethane (PU)-based stretchable iTE material, exhibiting $Y$ of 0.63 MPa, maximum strain of 156%, $S_i$ of 34.5 mV·K⁻¹, and $ZT_i$ of 1.3[37]. Xu et al. enhanced the maximum strain to 300% using a PU-based iTE material with a borate crosslinker ($Y$ of 0.79 MPa, $S_i$ of 34.5 mV·K⁻¹, and $ZT_i$ of 0.99)[38]. Jia et al. presented a stretchable and self-healable iTE material composed of an ionic liquid ([EMIm][Tf₂N]) with an amide-based self-healing polymer; however, they reported $S_i$ of only -1.4 mV·K⁻¹ and no $ZT_i$[39]. Recently, we reported stretchable and self-healable iTE materials ($S_i$ of 8.1–38.3 mV·K⁻¹ and $ZT_i$ of 1.04–2.34 at 90% relative humidity, RH)[40,41]. Very recently, Cho et al. reported a poly(vinyl alcohol) (PVA)-based self-healable iTE material with $ZT_i$ of 7.2 (at 80% RH)[36].

In this study, we developed an remarkable, highly performant stretchable/self-healable iTE material by optimizing the thermophoresis of protons in a polymer complex PEDOT:PAAMPSA:PA, poly(2-acrylamido-2-methyl-1-propanesulfonic acid) (PAAMPSA)- doped poly(3,4-ethylenedioxythiophene) (PEDOT) with phytic acid (PA). The PEDOT:PAAMPSA:PA achieved a $ZT_i$ of 12.3, $S_i$ of 21.9 mV·K⁻¹, $\sigma_i$ of 0.309 S·cm⁻¹, and $\kappa$ of 0.358 W·m⁻¹·K⁻¹ at 70% RH. Notably, the $ZT_i$ value was the highest among the iTE materials so far reported in the literature. Controlling the carrier (i.e., proton) concentration while minimizing the contribution of counter ions was the primary cause for the excellent iTE performance. Furthermore, PEDOT:PAAMPSA:PA achieved high stretchability and excellent self-healing ability. Notably,

the high $ZT_i$ was retained under various mechanical stresses (i.e., repeated self-healing and stretching). An ITEC device based on the PEDOT:PAAMPSA:PA demonstrated a maximum power output and energy density of 4.59 μW·m⁻² and 1.95 mJ·m⁻², respectively. Moreover, a 9-pair ITEC module, in which the PEDOT:PAAMPSA:PA (positive $S_i$) was paired with the reported NPC40 (negative $S_i$)[30] achieved a thermovoltage of 0.37 V·K⁻¹ with a maximum power output of 0.21 μW·m⁻² and energy density of 0.35 mJ·m⁻² at 80% RH.

## Results

### Synthesis of PEDOT:PAAMPSA:PA
The self-healable iTE material PEDOT:PAAMPSA:PA was synthesized by adding a monomer, 3,4-ethylenedioxythiophene (EDOT), to an aqueous solution of an anionic polymer (PAAMPSA) and physical cross-linker (PA). Separately, a solution of a chemical oxidant (ammonium persulfate, APS) in deionized water was prepared, and the two solutions were mixed to initiate polymerization followed by stirring in an ice bath under ambient air conditions. The oxidative polymerization of EDOT was indicated by the color change of the aqueous solution from light brown to dark blue (Fig. 1b). The complexes of positively charged PEDOT chains and negatively charged PAAMPSA chains were formed, which resembles the well-known PEDOT:PSS complex[42–44]. Fig. 1a schematically illustrates the molecular structure of PEDOT:PAAMPSA:PA. The polymerized PEDOT (blue strand) is aligned along the amorphous PAAMPSA (orange strand), whereas the multivalent PA (yellow sphere) is physically crosslinked through hydrogen bonds (pink dashed line) and electrostatic interactions (blue dashed line). The soft nature of PEDOT:PAAMPSA and the physical crosslinking network by PA resulted in stretchability and self-healing (Fig. 1a), which will be discussed later.

### Optimization of the iTE properties of PEDOT:PAAMPSA:PA
It is known that the $n_i$, $D_i$, and $S_i^*$ of iTE materials are crucial factors in determining their $S_i$ and $\sigma_i$. In PEDOT:PAAMPSA:PA, the primary charge carrier in the thermophoresis is the proton (H⁺) because the polymeric anions have a significantly slower diffusion compared to the small-sized proton. In the synthesis of PEDOT:PAAMPSA:PA, there are two primary sources of protons: the oxidative polymerization of EDOT wherein two equivalent protons are dissociated per EDOT-EDOT bonding (Supplementary Fig. 1)[45], and dissociation of PAAMPSA which can be expressed as,

$$RSO_3H \rightleftharpoons RSO_3^- + H^+ \qquad (5)$$

in an equilibrium state. After polymerization, the positively charged PEDOT chains and negatively charged PAAMPSA chains form aligned $RSO_3^- - PEDOT^+$ complexes (Supplementary Fig. 1), which facilitates the forward reaction in the PAAMPSA equilibrium according to Le Chatlier's principle[45,46]. Additionally, this acidic environment facilitates the formation of hydrogen sulfate anions ($HSO_4^-$) originating from the APS initiator (Supplementary Fig. 1)[36]. Therefore, we posited that the PEDOT/PAAMPSA ratio significantly influences the proton concentration in PEDOT:PAAMPSA:PA. To evaluate our hypothesis, the pH values of various PEDOT:PAAMPSA solutions were examined (Supplementary Table 1). As the PEDOT/PAMMPSA ratio increased, the pH of the PEDOT:PAAMPSA solution decreased, indicating a higher proton concentration. X-ray photoelectron spectroscopy (XPS) spectra of PEDOT:PAAMPSA films revealed consistent results. As the PEDOT/PAMMPSA ratio increased, the S 2p peak near 168 eV shifted to higher energy (Supplementary Fig. 2). This result indicates that the electron density of the sulfur atom in PAAMPSA was decreased by the $RSO_3^- - PEDOT^+$ interaction; thus, a higher photon energy was required to induce the photoelectric effect.

To determine the effects of the PEDOT/PAAMPSA ratio on the iTE properties, PEDOT:PAAMPSA:PA was deposited by spin-coating on two

lines of thermally evaporated Au electrodes on glass substrates (Supplementary Fig. 3). As shown in Fig. 2a, the $S_i$ and $\sigma_i$ values of the PEDOT:PAAMPSA:PA films were affected by the PEDOT/PAAMPSA ratio. The $S_i$ and $\sigma_i$ values increased for a PEDOT content of up to 6.2 wt.%, then decreased at higher values of the PEDOT content. The ionic power factor ($PF_i = S_i^2 \sigma_i$) was optimized for a PEDOT content of 6.2 wt.% to be 14.8 mW·m⁻¹·K⁻² ($S_i$ = 21.9 mV·K⁻¹, and $\sigma_i$ = 0.309 S·cm⁻¹) at the 70% RH (Fig. 2b). Meanwhile, the electronic conductivity was negligible regardless of PEDOT content (~9.5 × 10⁻⁴ S·cm⁻¹), indicating that the conductivity of our material is dominated by $\sigma_i$ (Supplementary Fig. 4, Supplementary Fig. 5, and Supplementary Fig. 6). Notably, PA negligibly influenced the iTE properties of the films (Supplementary Fig. 7), acting solely as a physical crosslinker. Figure 2c shows the changes in the net $n_i$ and $D_i$ of the PEDOT:PAAMPSA films with respect to the PEDOT/PAAMPSA ratio. Each net $n_i$ and $D_i$ is determined by fitting the dielectric constant-frequency curve (Supplementary Fig. 8). The net $n_i$ increased as the PEDOT/PAAMPSA ratio increased, whereas net $D_i$ decreased. Although $S_i$ is not directly determined by the value of $n_i$, the increase in $n_i$ influences the $S_i^*$ (i.e., the change in $\Delta S_{struc}$) of the carriers. When the PEDOT/PAAMPSA ratio increases, the proton concentration increases, resulting in a more kosmotropic environment for protons because of the reduced average distance between the protons[17,20]. As a result, the $S_i$ of PEDOT:PAAMPSA:PA continuously improved for a PEDOT content of up to 6.2 wt.%, owing to an increase in $n_i$ (Fig. 2d vs. e)[26].

However, when the PEDOT content is 9.1 wt.% or higher, the formation of PEDOT aggregates due to excess PEDOT results in a suboptimal contribution to the proton concentration. The formation of a linearly aligned PEDOT:PAAMPSA complex optimizes the proton dissociation from PAAMPSA; however, the PEDOT aggregates prevent efficient proton dissociation[42,47]. Simultaneously, the concentration of $HSO_4^-$ (anion) originating from the APS initiator increases proportionally with the PEDOT wt.% and begins to compete with the proton (cation) for the determination of the $S_i$ value (Fig. 2f).

To optimize the $S_i$ value, $n_i$ and $D_i$ of the primary charge carrier (proton in our case) should be dominant[14]. The PEDOT aggregates (dark color) in the films with higher PEDOT content (≥9.1 wt.%) were clearly observed in the optical microscopy (OM) and scanning electron microscopy (SEM) images of the PEDOT:PAAMPSA films (Fig. 3a, b). The formation of the PEDOT aggregates is also indicated by the net $D_i$ decrease, which hinders the thermo-diffusion of both ion carriers (Fig. 2c), as reported for the conventional PEDOT:PSS system[30,47]. Additionally, $\sigma_i$ of the monovalent ion system is determined by the following equation:

$$\sigma_i = n_i \cdot D_i \cdot \frac{e^2}{kT} \tag{6}$$

where $e$ is the elementary charge, and $k$ is the Boltzmann constant[48]. Differently from $S_i$, higher net $n_i$ and $D_i$ are required to achieve high $\sigma_i$, regardless of the type of carrier (cation or anion). At a 6.2 wt.% PEDOT content, $\sigma_i$ was optimized owing to the intermediate net $n_i$ and $D_i$. At lower PEDOT/PAAMPSA ratios, $n_i$ is too low owing to insufficient proton dissociation, although $D_i$ is higher because of negligible PEDOT aggregates. Conversely, at higher PEDOT/PAAMPSA ratios, $n_i$ is higher, but $D_i$ is lower, as discussed above (Fig. 2c).

The iTE properties of the PEDOT:PAAMPSA:PA films were enhanced at higher RH values. The $PF_i$ of the PEDOT:PAAMPSA:PA film at RH of 90% was improved to 60.8 mW·m⁻¹·K⁻² ($S_i$ = 27.3 mV·K⁻¹, and $\sigma_i$ = 0.816 S·cm⁻¹, Fig. 2b and Supplementary Fig. 9). There are two essential functions of water absorption in systems: the formation of ion transport channels and the facilitation of charge dissociation[26]. Under sufficient RH conditions, the water molecules form channels between the polymer chains, acting as a highway that enhances $D_i$[25]. Furthermore, the water molecules facilitate charge dissociation from the acid groups owing to their high polarity, which increases $n_i$. The effects of water molecules on the $D_i$ and $n_i$ of the PEDOT:PAAMPSA:PA film are confirmed in Supplementary Fig. 10.

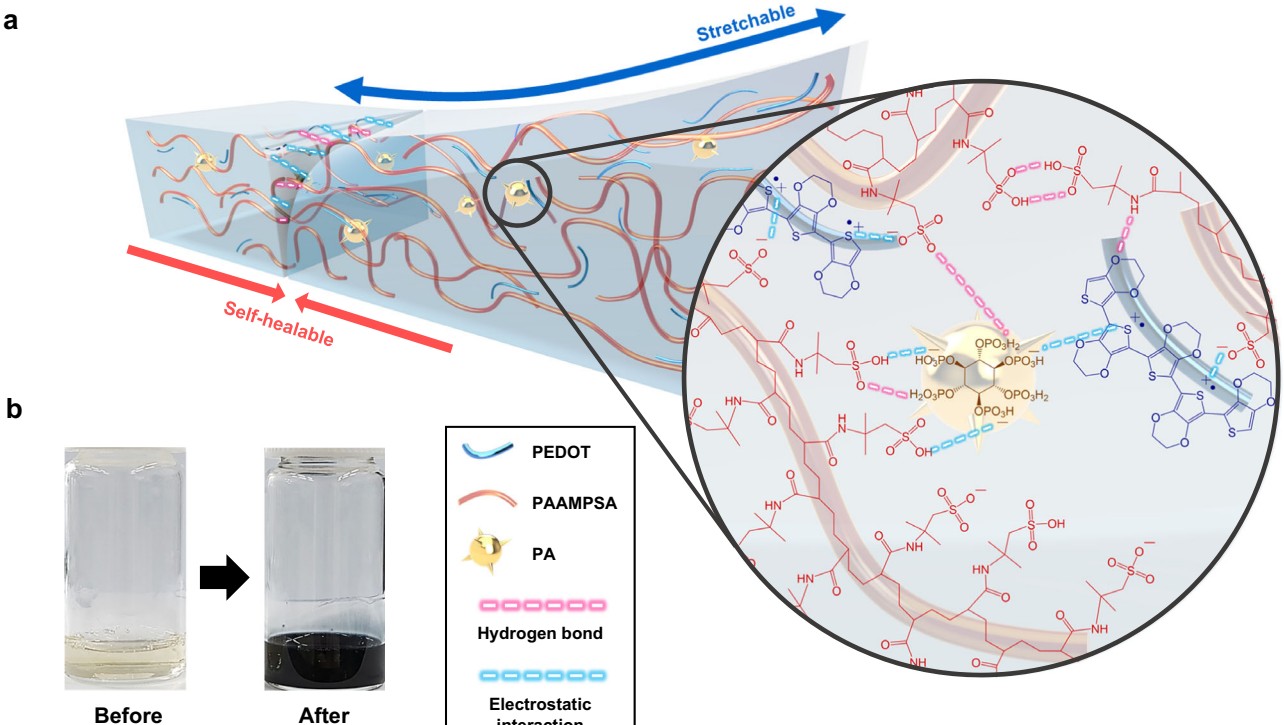

**Fig. 1 | Synthesis and morphology of a PEDOT:PAAMPSA:PA polymer complex. a** Schematic illustration of the molecular structure and mechanical properties (stretchability and self-healability) of the PEDOT:PAAMPSA:PA film, **b** Photos of aqueous solutions before and after the PEDOT:PAAMPSA:PA polymerization.

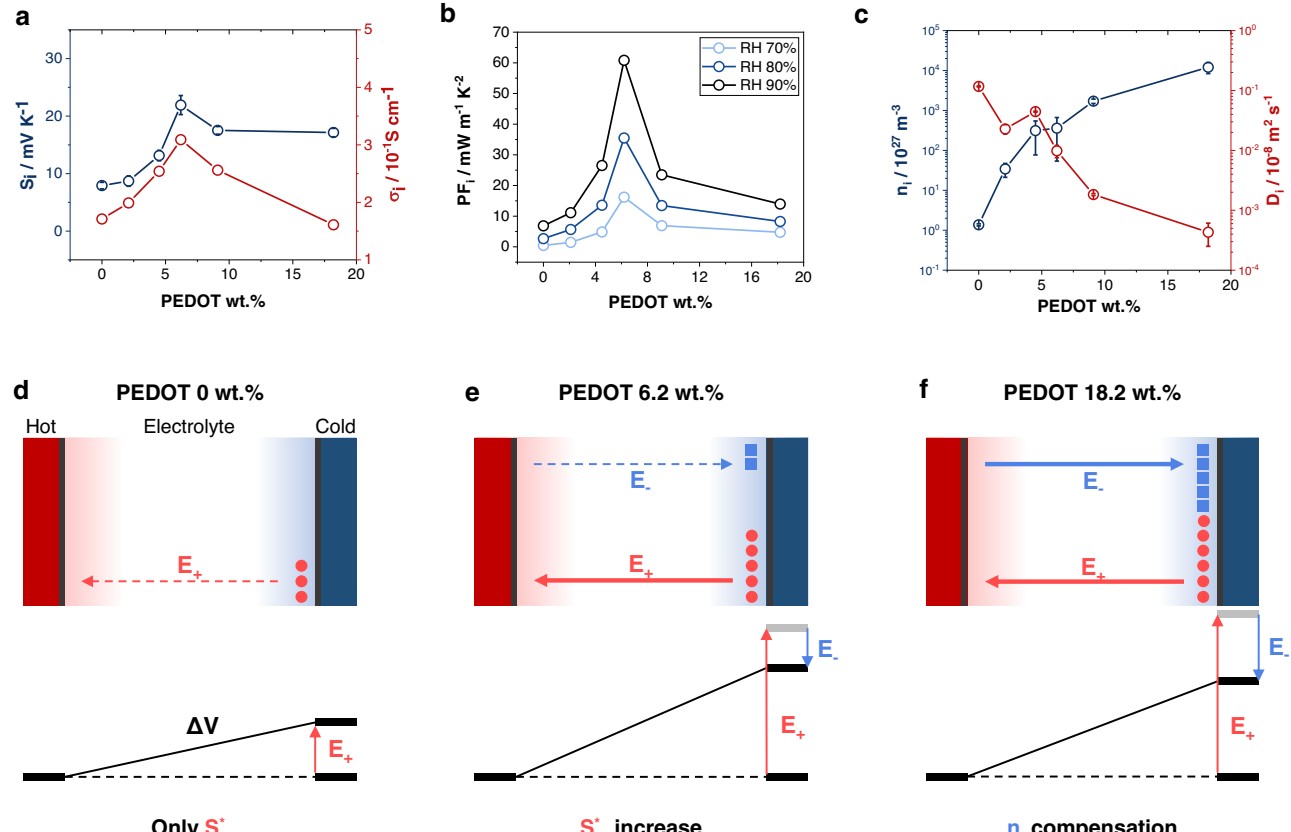

**Fig. 2 | iTE properties of PEDOT:PAAMPSA:PA films. a** $S_i$ and $\sigma_i$ of PED-OT:PAAMPSA:PA with different PEDOT/PAAMPSA ratios at 70% RH. **b** $PF_i$ of the PEDOT:PAAMPSA:PA at different RH. **c** Net $n_i$ and $D_i$ of PEDOT:PAAMPSA with different PEDOT/PAAMPSA ratios at 80% RH. Each $S_i$ value in (**a**) is averaged from

five samples and each $n_i$ and $D_i$ in (**c**) is averaged from three samples. **d**–**f** Schematic illustration of the effects of the internal environment on the $S_i$ of PEDOT:PAAMPSA:PA. The subscript "+" indicates a cation (proton in this work) and "−" indicates an anion ($HSO_4^-$ in this work).

To determine the $ZT_i$ of the PEDOT:PAAMPSA:PA film, its thermal conductivity ($\kappa$) is determined by the following equation:

$$\kappa = C_p \cdot \rho \cdot \alpha \tag{7}$$

where $C_p$ is the specific heat capacity, $\rho$ is the density of the sample, and $\alpha$ is the thermal diffusivity. Laser flash analysis (LFA) and differential scanning calorimetry (DSC) were performed to obtain $\alpha$ and $C_p$, respectively (see method for detail). The $\kappa$ value of the PED-OT:PAAMPSA:PA film (6.2 wt.% PEDOT content) at 70% and 90% RH were calculated to be 0.358 and 0.403 W·m⁻¹·K⁻¹, respectively (Supplementary Fig. 11). Based on this result, the $ZT_i$ of the optimized PED-OT:PAAMPSA:PA film was determined to be 12.3 and 44.9 at 70% and 90% RH, respectively (Table 1)[8]. To the best of our knowledge, these values are the highest $ZT_i$ among all previously reported iTE materials, including non-stretchable and non-self-healable ones (Fig. 3c and Supplementary Table 2)[36].

**Mechanical properties of PEDOT:PAAMPSA:PA**
In addition to the significantly high $ZT_i$ value, the PED-OT:PAAMPSA:PA films were highly stretchable and self-healable under ambient conditions. This self-repairing property originates from the dynamic interactions between PEDOT, PAAMPSA, and PA, wherein the dispersed PAs physically crosslink the PEDOT:PAAMPSA complexes by hydrogen bonds and electrostatic interactions (Fig. 1a)[49]. As shown in Fig. 4a, the free-standing films stretched up to >1000% from their initial state. This result is the highest stretchability among the reported iTE materials in the literature[37,38,50]. Reproducible mechanical stretchability (a strain of >500%) was easily

demonstrated in the PEDOT:PAAMPSA:PA free-standing film (Supplementary Movie 1). Self-healing characteristics were demonstrated using two types of samples: a thin film on a glass substrate and a free-standing film. The PEDOT:PAAMPSA:PA thin film on a glass substrate was perfectly self-healed over a period of 2 h without any external stimuli, even at 50% RH. The self-healing process is significantly accelerated under higher RH conditions; for example, self-healing was completed in 60 s at 70% RH (Fig. 4b, Supplementary Fig. 12, and Supplementary Movie 2)[51,52]. During the self-healing process, there was no microscopic elemental change in the composition of the PEDOT:PAAMPSA:PA sample, as indicated by the energy dispersive spectroscopy (EDS) mapping results (Supplementary Fig. 13).

Figure 4c shows the stress-strain characteristics of the PED-OT:PAAMPSA:PA free-standing films. The maximum stress in the films decreased for the film with 2.1 wt.% PEDOT and consistently increased with the increase in the PEDOT content, demonstrating 450 kPa at 6.2 wt.% PEDOT[53]. The average maximum strain and toughness of the PEDOT:PAAMPSA:PA free-standing films were also enhanced as the PEDOT/PAAMPSA ratio increased (Fig. 4d), owing to the more vital dynamic interaction among the components. To investigate the effects of the PEDOT/PAAMPSA ratio on the self-healing behavior of the PEDOT:PAAMPSA:PA free-standing films, the films with various PEDOT/PAAMPSA ratios were cut and attached without any external stimuli at 70% RH. A free-standing film with 2.1 wt.% PEDOT was healed after 6 h, that with 6.2 wt.% PEDOT was healed after 2 h, whereas that with 18.2 wt.% PEDOT was completely healed after only 0.5 h. The perfect self-healing of the PEDOT:PAAMPSA:PA free-standing film is shown in Supplementary Movie 3. Notably, the mechanical properties of the PEDOT:PAAMPSA:PA film were enhanced as the

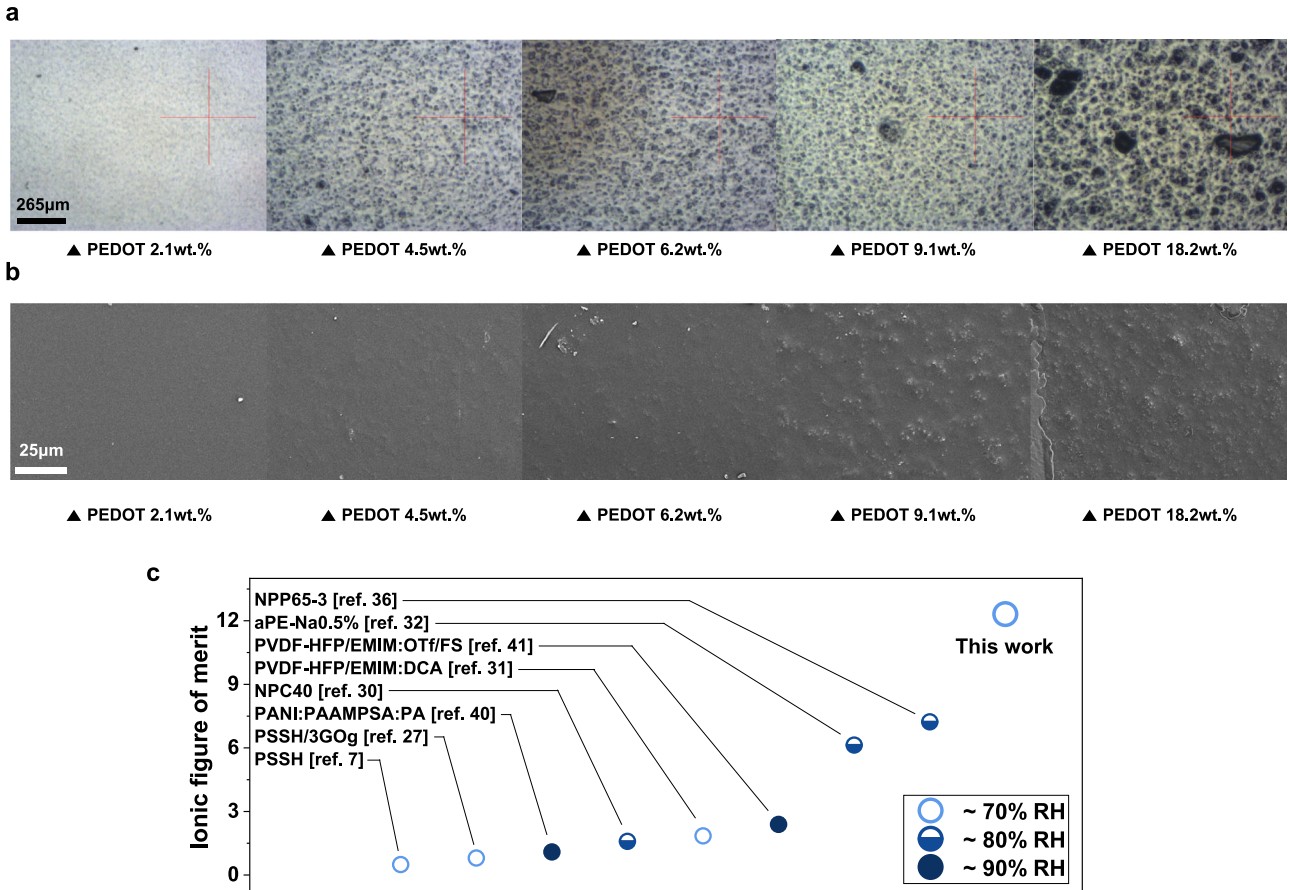

**Fig. 3 | Morphology of PEDOT:PAAMPSA and $ZT_i$ of the optimized PED-OT:PAAMPSA:PA film. a** OM and **b** SEM images (×1000 magnification) of the PEDOT:PAAMPSA films with various PEDOT/PAAMPSA ratios. **c** Summary of the $ZT_i$ values of the reported state-of-the-art iTE materials in the literature. The $ZT_i$ of the reported iTE materials was measured at various RH values (from 70–90%).

PEDOT/PAAMPSA ratio increased because of stronger intermolecular interactions, whereas its iTE properties were optimized at a 6.2 wt.% PEDOT content due to the optimized balance between net $n_i$ and $D_i$ values.

Most intriguingly, the iTE properties of the PEDOT:PAAMPSA:PA film were perfectly retained after various mechanical stresses (self-healing and stretching). Even after multiple self-healing steps (cut and repair) during ordinary $S_i$ measurement, the $S_i$ was almost recovered (Fig. 4e). A period of <60 s was required to recover $\sigma_i$ at 70% RH, which is consistent with the self-healing time of a thin film on a glass substrate (Supplementary Fig. 14). The $S_i$ and $\sigma_i$ of the films retained 95% and 90% of their original values after 20 and 30 times of self-healing cycles, respectively (Fig. 4f). A similar trend was observed in repeated stretching cycle tests. The $S_i$ values were

retained at >95% of the original values after 50 stretch-release cycles (a strain of 100%) (Fig. 4g). These results revealed that the microscopic proton pathway was perfectly intact even after multiple self-healing and stretching cycles, resulting in the retention of the iTE performance.

### iTE performance of the ITEC device and module

In view of the excellent iTE and mechanical properties, we fabricated an ITEC device using PEDOT:PAAMPSA:PA films. Figure 5a represents the operating principle of the ITEC, and Fig. 5b shows the thermovoltage profile during the operation of the ITEC[9]. First, the heat flux from the hot side to the cold side induces the thermo-diffusion of the proton by the Soret effect (Stage I). Subsequently, when the load resistor is connected, the electrons flow through an external circuit due to thermovoltage (Stage II). Because the accumulated protons cannot move through the external circuit, the electrons remain on the cold-side electrode. When the thermovoltage is canceled, the load resistor is disconnected, the temperature gradient is removed, and the accumulated protons return to their original site (Stage III). In this state, a negative open-circuit voltage is built up due to the remaining electrons on the electrode. Finally, when the load resistor is connected again, the electrons flow in the opposite direction through the external circuit (Stage IV). To investigate the energy-storage capacity of the PEDOT:PAAMPSA:PA ITEC device, the maximum power output and energy density with different load resistors were measured (Fig. 5c and Table 2). Both power output and energy density were optimized to be 4.59 μW·m⁻² and 1.95 mJ·m⁻², respectively, at a load resistance of 10 kΩ.

**Table 1 | iTE properties of PEDOT:PAAMPSA:PA films with different PEDOT/PAAMPSA ratios at 70% RH**

| PEDOT wt.% | $S_i$ / mV K⁻¹ | $\sigma_i$ / S cm⁻¹ | $PF_i$ / mW m⁻¹ K⁻² | $\kappa$ / W m⁻¹ K⁻¹ | $ZT_i$ |
|---|---|---|---|---|---|
| 0 | 7.91 ± 0.57 | 0.171 | 1.18 | | |
| 2.1 | 8.70 ± 0.90 | 0.199 | 1.50 | | |
| 4.5 | 13.1 ± 0.91 | 0.254 | 4.39 | | |
| 6.2 | 21.9 ± 1.66 | 0.309 | 14.8 | 0.358 | 12.3 |
| 9.1 | 17.5 ± 0.66 | 0.256 | 7.87 | | |
| 18.2 | 17.1 ± 0.73 | 0.161 | 4.76 | | |

The ITEC performance was identical before and after repeated self-healing cycles, owing to the excellent self-healing ability of PEDOT:PAAMPSA:PA (Fig. 5d).

To demonstrate the potential of ITEC, we fabricated an ITEC module composed of 9-pairs of legs[30]. Since most iTE materials produce tens of mV·K⁻¹, which is still far from practical applications, voltage boosting by series connection would be a compelling strategy. The legs of the ITEC module are alternately connected to PEDOT:PAAMPSA:PA (positive $S_i$ legs) with NPC40 (negative $S_i$ legs)[30] on a flexible polypropylene (PP) substrate with thermally deposited gold electrodes (Fig. 5e). The ITEC module with 9-pair legs demonstrated ~0.37 V·K⁻¹ at 80% RH, which is consistent with the addition of the $S_i$ value of positive and negative $S_i$ legs (Fig. 5f). Furthermore, the operation of the ITEC module was successful, indicating that the two different iTE materials could be combined into a single device (Supplementary Fig. 15). The maximum power output and energy density of the ITEC module were 0.21 μW·m⁻² and 0.35 mJ·m⁻², respectively (Supplementary Fig. 16). This result demonstrates the potential of the high-voltage ITEC module, which supports the use of our ITEC module as a wearable power source.

## Discussion

Our results demonstrate that the optimization of the balance of net $n_i$ and $D_i$ is crucial for achieving high-performance stretchable and self-healable iTE materials. The PEDOT/PAAMPSA ratio was found to be a key factor in determining the iTE properties of the PEDOT:PAAMPSA:PA film, with higher ratios resulting in improved iTE properties due to the higher proton concentration. However, excessive PEDOT content deteriorated the iTE properties by forming PEDOT aggregates. Our findings suggest that the dynamic interactions between the flexible PEDOT:PAAMPSA complex and multivalent physical crosslinker (PA) enable the highest stretchability compared to other state-of-the-art iTE materials.

Importantly, our PEDOT:PAAMPSA:PA film showed exceptional iTE properties, with a $ZT_i$ value of 12.3 at 70% RH, which is 70% higher than the previous record. Our film also retained its iTE properties even after multiple self-healing and stretching cycles, demonstrating the potential for long-term use in practical applications. The high power output and energy density of our iTEC device and ITEC module with 9-pair legs suggest that our material has potential for use as a self-power source for wearable electronic devices.

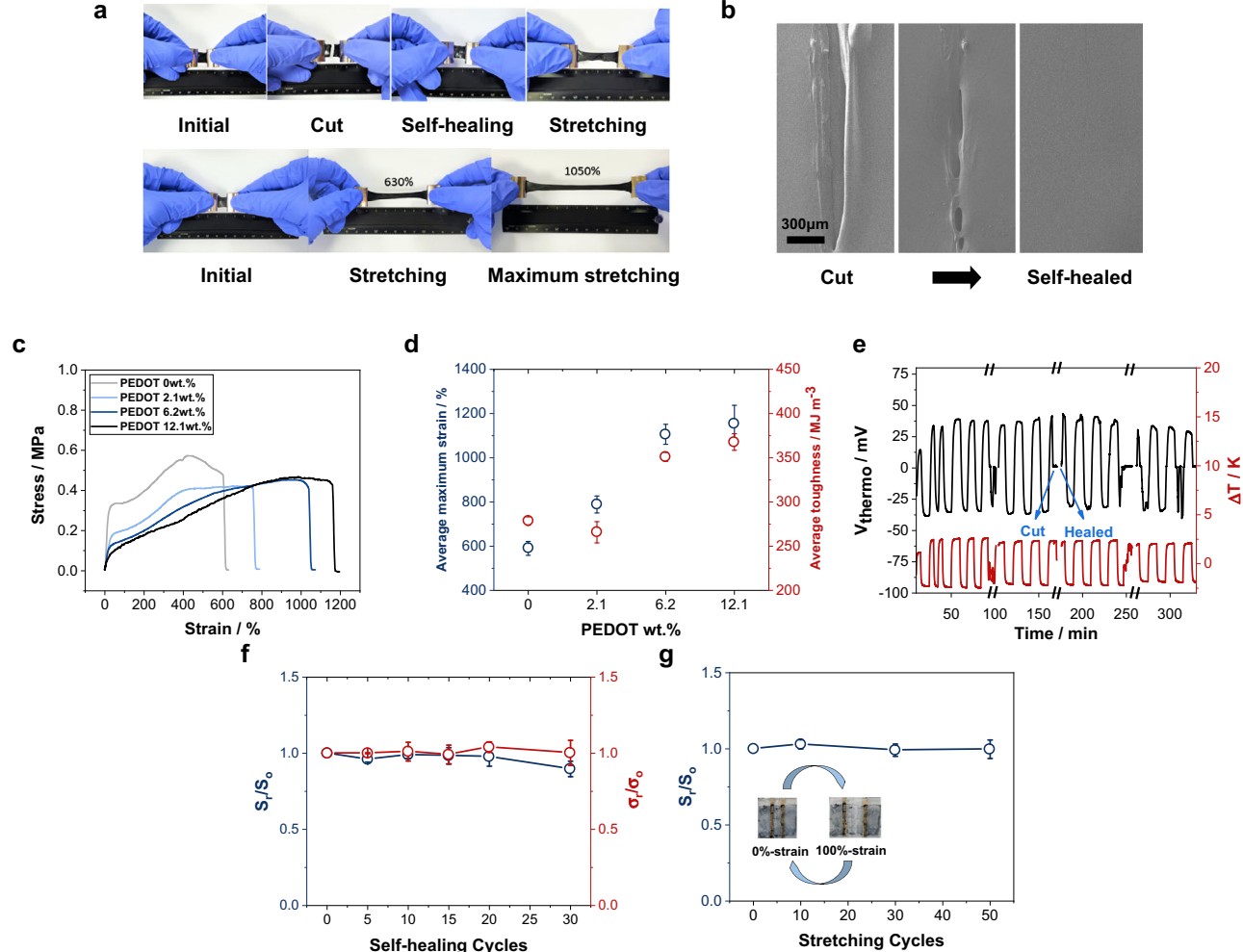

**Fig. 4 | Mechanical properties of PEDOT:PAAMPSA:PA films. a** Photos of the self-healing and stretching of a PEDOT:PAAMPSA:PA (6.2 wt.% PEDOT) free-standing film. **b** SEM images of each self-healing stage of a PEDOT:PAAMPSA:PA (6.2 wt.% PEDOT) thin film on a glass substrate at 70% RH. Scale bar indicates 300 μm. **c** Stress-strain curves of PAAMPSA:PA and PEDOT:PAAMPSA:PA free-standing films with different PEDOT/PAAMPSA ratios at 60% RH. The thickness of the free-standing films is 0.8 mm. **d** Average maximum strain and toughness of PAAMPSA:PA and PEDOT:PAAMPSA:PA free-standing films with different PEDOT/ PAAMPSA ratios at 60% RH. **e** Thermovoltage profile of a PEDOT:PAAMPSA:PA (6.2 wt.% PEDOT) thin film during a real-time self-healing at 70% RH. **f** Stability of $S_i$ and $\sigma_i$ during repeated self-healing cycles at 70% RH. **g** Stability of $S_i$ during repeated stretching cycles (strain of 100%) at 70% RH. The subscript "o" indicates the original values before mechanical stresses and "r" indicates the recovered values after mechanical stresses. All values in (**d**), (**f**), and (**g**) are averaged from three samples each.

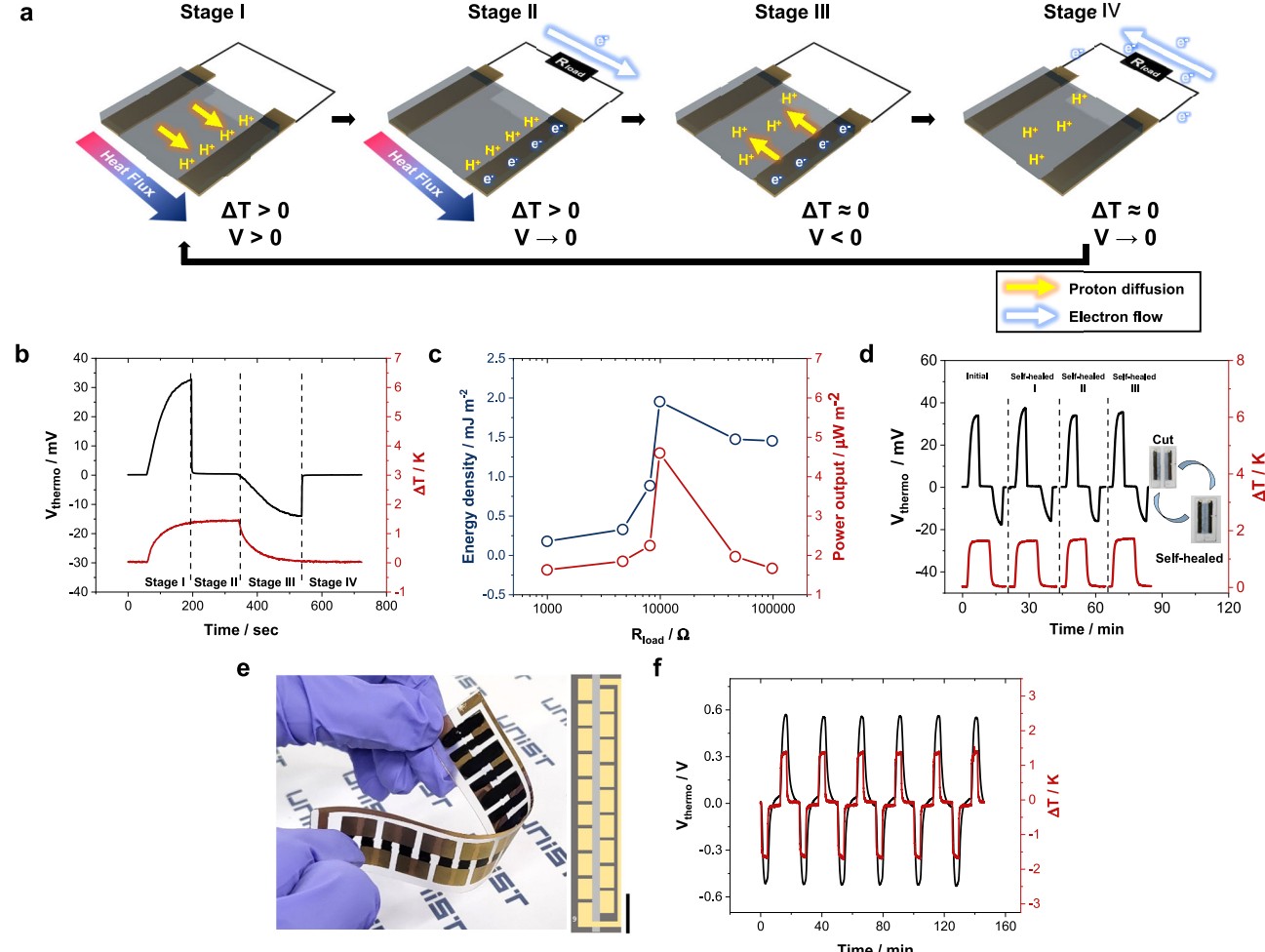

Fig. 5 | iTE properties of the PEDOT:PAAMPSA:PA ITEC device and ITEC module composed of PEDOT:PAAMPSA:PA (positive $S_i$ material) and NPC40 (negative $S_i$ material). a Schematic illustration of the operation of ITEC. b Thermovoltage profile of PEDOT:PAAMPSA:PA ITEC device during ITEC operation at 70% RH. A 10 kΩ load resistor was connected for charging and discharging in stage II and IV. c Energy density and power output of the PEDOT:PAAMPSA:PA ITEC device with different load resistors at 70% RH. d Thermovoltage profile of PEDOT:PAAMPSA:PA ITEC device during real-time self-healing cycle at 70% RH. e Photo of ITEC module. Scale bar indicates 2 cm. f Reproducible thermovoltage profile of the ITEC module at 80% RH.

## Methods

### Synthesis of PEDOT:PAAMPSA:PA and PEDOT:PAAMPSA complexes

To prepare a PEDOT:PAAMPSA:PA (6.2 wt.% PEDOT content) solution, 10 g of a PAAMPSA aqueous solution (15 wt.%), 0.13 g of EDOT, and 0.97 g of a PA aqueous solution (50 wt.%) were mixed and slowly stirred in an ice bath. Simultaneously, 0.18 g of APS powder was mixed with 1 mL of deionized water to prepare an APS aqueous solution. The molar ratio of EDOT to APS was 0.85. After stirring for 0.5 h, the APS solution was dropped into the EDOT:PAAMPSA:PA mixture to initiate the polymerization of EDOT. The completed PEDOT:PAAMPSA:PA solution could be used for 24 h. The PEDOT:PAAMPSA solution was also synthesized using the same procedure, but without PA.

### Preparation of the ITEC device

PEDOT:PAAMPSA:PA and PEDOT:PAAMPSA ITEC devices were fabricated and their iTE properties were investigated. First, a glass substrate was treated in a UV-$O_3$ cleaner (AHTECH, AC-6) for 30 min. Two lines of gold electrode (thickness:80 nm) were deposited on the glass at reduced pressure (<$10^{-6}$ Torr) using a thermal evaporator. The ITEC device was fabricated by spin-coating PEDOT:PAAMPSA:PA and PEDOT:PAAMPSA solutions on the glass substrate (2000 RPM, 30 s). The films were then heated for 5 min at 50 °C. The thickness of the sample was ~5 μm, as determined using a surface profilometer (KLA Tencor P6).

### Synthesis of NPC40

0.01 g of the $CuCl_2$ powder was mixed with 1 g of the PEDOT:PSS solution. The weight ratio of $CuCl_2$ to PEDOT:PSS was 0.4. The solution was vortexed by a mixer (Scientific Industries, SI-0256) for 10 min.

**Table 2 | Summary of the energy density and power output of the PEDOT:PAAMPSA:PA (6.2 wt.% PEDOT) ITEC device with different load resistors at 70% RH**

| $R_{load}$ / ×10³ ohm | Energy / nJ | Charging time / s | Energy density / mJ m⁻² | Power output / μW m⁻² |
|---|---|---|---|---|
| 0.1 | 0.02 | 3.83 | 0.00 | 0.05 |
| 1 | 18.17 | 107.25 | 0.17 | 1.61 |
| 4.7 | 48.60 | 302.12 | 0.32 | 1.83 |
| 8.2 | 92.17 | 394.23 | 0.88 | 2.23 |
| 10 | 204.75 | 425.29 | 1.95 | 4.59 |
| 47 | 153.89 | 751.88 | 1.47 | 1.95 |
| 100 | 152.19 | 875.83 | 1.45 | 1.65 |

## Preparation of the ITEC module

A flexible PP substrate (2.5 × 1 cm) was treated in a UV-O$_3$ cleaner (AHTECH, AC-6) for 30 min. A designed gold electrode was deposited on the PP substrate using the same procedure as that used for the glass substrate (Fig. 5e). The gold-deposited PP substrate was again treated in a UV-O$_3$ cleaner for 10 min to form an uniform hydrophilic surface. Our ITEC module was fabricated by drop-casting a PEDOT:PAAMPSA:PA (positive $S_i$ material) solution and an NPC40 (negative $S_i$ material) solution on the substrate mentioned above. Each material was alternately drop-casted onto the substrate to realize a synergic thermovoltage. The films were then dried under ambient conditions (60% RH, 23–25 °C) for 3 h.

## Data availability

The authors declare that the data supporting the findings of this study are available within the paper and its supplementary information files. All other additional data are available from the corresponding author upon request. Source data are provided with this paper.

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

## Acknowledgements

The authors gratefully acknowledge the support from the National Research Foundation of Korea (NRF) grant funded by the Korean government (MSIP, Grant Nos. 2023R1A2C3002881, 2022M3H4A1A03076652, 2022R1A2C1092273, and 2019R1A2C2087218).

## Author contributions

D.-H.K., S.-Y.J., and J.-W.J. conceived the idea and designed the research. D.-H.K., Z.A.A., and Y.T.M. performed experiments and analyzed data. D.-H.K. analyzed the dielectric constant, performed pH, XPS, SEM, and OM measurements, and produced the ITEC module. Z.A.A. and Y.T.M. measured the mechanical properties and stability of thermoelectric properties. S.-Y.J. and J.-W.J. supervised the project. All authors contributed to the writing of the paper.

## Competing interests

The authors declare no competing interests.
