## [Peer Review File · Nature Communications]

Self-healable Polymer Complex with a Giant Ionic Thermoelectric EffectReviewers' Comments:

Reviewer #1:

Remarks to the Author:

The manuscript by Dong-Hu Kim et al. describes a synthesis of PEDOT derivative, with a very favorable mechanical properties and ionic thermoelectric effect. I can comment mostly on the synthesis and mechanical properties evaluation. The manuscript is well written and the story line is fluent and clear to the reader. In addition, the thorough explanation regarding the reaction mechanism is very important for the scientific community and helps support the finding of the paper. The noteworthy results are that the ZTi value was very high among the iTE materials, and the very high stretchability and toughness of the material.

I recommend this work for publication after reviewing the following issues:

1. Supplementary Fig. 6. There are no error bars/average/SD. In addition, is it possible to measure more humidity percentages? To straighten the claim?
2. Fig 4: Stress-strain curves were done in 60% RH humidity. Why did the authors choose this percentage? This percentage is not shown in the manuscript. Figure 4d should be better represented in a bar graph, with error bars, and SD, as the average of at least 3 measurements. Showing a line connecting the dots might be misleading since the measurement was not done for all the PEDOT concentrations shown on the X axis. Figures 4F and 4g should also contain error bars and indicate the number of repetitions from which the average was calculated. The authors should indicate the thickness of the fabricated free-standing films.
3. The authors claim: "Stretchability exceeding that of the human skin (a strain of >500%) was easily demonstrated in the PEDOT:PAAMPSA:PA free-standing film (Supplementary Movie 1)." Can the authors repeat figure 4e with challenging up to 500% strain? This is about 50 % of the max strain achieved, and a relevant result for skin stretchability. However, if it does not generate stable cycles, please rephrase this statement.
4. The discussion summarizes the result section. The authors should benchmark their results against other reported iTE materials, compare the mechanical performance of other similar formulations in the literature, or the electrical properties of the materials they used in other publications and also when demonstrated in a different application.

Reviewer #2:

Remarks to the Author:

This is a very interesting and well-written manuscript dealing with stretchable and self healable devices based on a polymer complex, including the conducting polymer PEDOT.

On my opinion the manuscript can be a potential candidate for publication in Nature Communications, after addressing the following issues.

- 1) The title talks about "Giant Ionic Thermoelectric Effect". How can the adjective "Giant" be justified? Maybe a table to compare state of the art results would be helpful.
- 2) The authors cite a stretchability of 500% for human skin, which looks a bit exaggerated. I believe a reasonable value would be around 30%.
- 3) Figure 2: the caption says that the values in Figure an and C are averaged on 5 and 3 samples respectively. However, no error bars are shown.
- 4) Figure 3b legend are invisible. Suggest to remove them and leave the scale bar only.

- 5) The statistics about the mechanical measurements (stress-strain) is unclear. How many samples have been measured to extract the stretchability values? Please explain the meaning of maximum strain and toughness in Figure 4d and explain how the values were obtained.
- 6) Please give more details about the self healing experiment. How exactly the cuts were performed (e.g. in figure 2b)? How was humidity controlled during the experiment? Would it be possible to define a healing efficiency (from Fig 4g it looks close to 100%)
- 7) Although it might not be relevant for this work, do the authors know about the electrical conductivity of the materials?
- 8) I believe that for a such high-level publication, the healing mechanism needs to be better described, although I understand it might not be straightforward. Anyway the authors could discuss the role of the different components of the blend in the self-healing process and try to explain why the healing time seems to strongly depend on the humidity.
- 9) The caption of Fig 4b is a bit misleading since it would mean that the SEM image is taken during the healing process. Please clarify.
- 10) Table 1: please explain how the error is evaluated.
- 11) The following literature about self healing PEDOT-based materials should be considered.

Y. Li, X. Zhou, B. Sarkar, N. Gagnon-Lafrenais, and F. Cicoira, Recent progresses on Self-healable Conducting Polymers, *Advanced materials*, 2108932, 2022.

X. Zhou, A. Rajeev, A. Subramanian, Y. Li, N. Rossetti, G. Natale, G. A. Lodygensky, and F. Cicoira, Self-healing, stretchable, and highly adhesive hydrogels for epidermal patch electrodes, *Acta Biomaterialia*, 139, 296-306, 2022.

Y. Li, S. Zhang, N. Hamad, K. Kim, L. Liu, M. Lerond, F. Cicoira, Tailoring the Self-Healing Properties of Conducting Polymer Films, *Macromol. Biosci.*, i2000146, 2020.

Y. Li, X. Li, S. Zhang, L. Liu, N. Hamad, S. R. Bobbara, D. Pasini, F. Cicoira, Autonomic Self-Healing of PEDOT:PSS Achieved via Polyethylene Glycol Addition, *Adv. Funct. Mater.*, 30, 2002853, 2020.

S. Zhang, F. Cicoira, Water-Enabled Healing of Conducting Polymer Films, *Adv. Mater.*, 29, 1703098, 2017.

Overall, I suggest publication after major revisions.

Point-by-point Response to Reviewer's Comments

REVIEWER COMMENTS

Reviewer #1 (Remarks to the Author):

The manuscript by Dong-Hu Kim et al. describes a synthesis of PEDOT derivative, with a very favorable mechanical properties and ionic thermoelectric effect. I can comment mostly on the synthesis and mechanical properties evaluation. The manuscript is well written and the story line is fluent and clear to the reader. In addition, the thorough explanation regarding the reaction mechanism is very important for the scientific community and helps support the finding of the paper.

The noteworthy results are that the ZT_i value was very high among the iTE materials, and the very high stretchability and toughness of the material.

I recommend this work for publication after reviewing the following issues:

1. Supplementary Fig. 6. There are no error bars/average/SD. In addition, is it possible to measure more humidity percentages? To straighten the claim?

Response: We appreciate the reviewer's comments. As the reviewer suggested, we measured the n_i and D_i values at various humidity conditions with error bars/average/SD. We also revised Supplementary Fig. 6 (changed to Supplementary Fig. 7) based on those results.

Revised Supplementary Fig. 7 | Net n_i and D_i of PEDOT:PAAMPSA:PA (6.2wt.% PEDOT) film at different RH values.

2. Fig 4: Stress-strain curves were done in 60% RH humidity. Why did the authors choose this percentage? This percentage is not shown in the manuscript.

Response: We appreciate the valuable feedback provided by the reviewer. Regarding the stress-strain curve measurement, we attempted to maintain the RH at 70%, but due to practical

constraints, we could only achieve a maximum of 60% in the measurement room. However, we believe that this RH level is sufficient to determine the general mechanical properties of the PEDOT:PAAMPSA:PA. Initially, the ambient humidity in the measurement room was below 40% RH, and we utilized an external humidity supply to raise it to 60%. We would like to request the reviewer's understanding of the limitations in our ability to control the humidity under these measurement conditions.

Figure 4d should be better represented in a bar graph, with error bars, and SD, as the average of at least 3 measurements. Showing a line connecting the dots might be misleading since the measurement was not done for all the PEDOT concentrations shown on the X axis.

Response: We would like to express our appreciation for the valuable comments provided by the reviewer. Based on the suggestion, we have improved our methodology by averaging the maximum strain and toughness values from three samples each. Additionally, we have made revisions to Fig. 4d by removing connecting lines and including error bars/average/SD to enhance the clarity of the data.

Revised Fig. 4c and 4d

Revised caption in Fig. 4d

(d) **Average** maximum strain and toughness of PAAMPSA:PA and PEDOT:PAAMPSA:PA free-standing films with different PEDOT/PAAMPSA ratios at 60% RH.

Revised manuscript

Fig. 4c shows the stress-strain characteristics of the PEDOT:PAAMPSA:PA free-standing films. The maximum stress in the films decreased for the film with 2.1 wt. % PEDOT and consistently increased with the increase in the PEDOT content, demonstrating 450 kPa at 6.2 wt.% PEDOT. The **average** maximum strain and toughness of the PEDOT:PAAMPSA:PA free-standing films were also enhanced as the PEDOT/PAAMPSA ratio increased (Fig. 4d), owing to the more vital dynamic interaction among the components.

The raw data for the three measurements are shown below.

Figures 4f and 4g should also contain error bars and indicate the number of repetitions from which the average was calculated. The authors should indicate the thickness of the fabricated free-standing films.

Response: We appreciate the reviewer’s comments. As the reviewer suggested, we revised Fig. 4f and 4g with error bars. The thickness of the free-standing film was 0.8 mm. We added this information to the caption of Fig. 4.

Revised Fig. 4f, 4g, and caption of Fig. 4

Fig. 4 | Mechanical properties of PEDOT:PAAMPSA:PA films. (a) Photos of the self-healing and stretching of a PEDOT:PAAMPSA:PA (6.2wt.% PEDOT) free-standing film. (b) SEM images of each self-healing stage of a PEDOT:PAAMPSA:PA (6.2wt.% PEDOT) thin film on a glass substrate at 70% RH. Scale bar indicates 300 μm . (c) Stress-strain curves of PAAMPSA:PA and PEDOT:PAAMPSA:PA free-standing films with different

PEDOT/PAAMPSA ratios at 60% RH. The thickness of the free-standing films is 0.8mm. (d) Maximum strain and toughness of PAAMPSA:PA and PEDOT:PAAMPSA:PA free-standing films with different PEDOT/PAAMPSA ratios at 60% RH. (e) Thermovoltage profile of a PEDOT:PAAMPSA:PA (6.2 wt.% PEDOT) thin film during a real-time self-healing at 70% RH. (f) Stability of S_i and σ_i during repeated self-healing cycles at 70% RH. (g) Stability of S_i during repeated stretching cycles (strain of 100%) at 70% RH. The subscript “o” indicates the original values before mechanical stresses and “r” indicates the recovered values after mechanical stresses. All values in (d), (f), and (g) are averaged from three samples each.

3. The authors claim: “Stretchability exceeding that of the human skin (a strain of >500%) was easily demonstrated in the PEDOT:PAAMPSA:PA free-standing film (Supplementary Movie 1).”

Can the authors repeat figure 4e with challenging up to 500% strain? This is about 50 % of the max strain achieved, and a relevant result for skin stretchability. However, if it does not generate stable cycles, please rephrase this statement.

Response: We appreciate the reviewer’s comments. Supplementary Movie 1 shows a stretch-release test demonstrating the reproducible “mechanical stretchability” of our material (up to ~500%). Additionally, Fig. 4e displays the maintenance of thermovoltage before and after the "self-healing" test, while Fig. 4g illustrates the maintenance of thermovoltage before and after the "100% stretching" cycle. To avoid any misunderstanding, we have revised the manuscript accordingly.

Revised manuscript

Reproducible mechanical stretchability (a strain of >500%) was easily demonstrated in the PEDOT:PAAMPSA:PA free-standing film (Supplementary Movie 1).

4. The discussion summarizes the result section. The authors should benchmark their results against other reported iTE materials, compare the mechanical performance of other similar formulations in the literature, or the electrical properties of the materials they used in other publications and also when demonstrated in a different application.

Response: We appreciate the valuable feedback provided by the reviewer. Based on the suggestion, we revised the discussion section. Additionally, we have added Supplementary Table 2, which compares both electric and mechanical properties between state-of-the-art iTE material in literature and ours. We expect this table could be helpful for readers to understand the excellence of our material.

Revised discussion

Our results demonstrate that the optimization of the balance of net n_i and D_i is crucial for achieving extraordinary high-performance stretchable and self-healable iTE materials. The PEDOT/PAAMPSA ratio was found to be a key factor in determining the iTE properties of the PEDOT:PAAMPSA:PA film, with higher ratios resulting in improved iTE

properties due to the higher proton concentration. However, excessive PEDOT content deteriorated the iTE properties by forming PEDOT aggregates. Our findings suggest that the dynamic interactions between the flexible PEDOT:PAAMPSA complex and multivalent physical crosslinker (PA) enable the highest stretchability compared to other state-of-the-art iTE materials.

Importantly, our PEDOT:PAAMPSA:PA film showed exceptional iTE properties, with a ZT_i value of 12.3 at 70% RH, which is 70% higher than the previous record. Our film also retained its iTE properties even after multiple self-healing and stretching cycles, demonstrating the potential for long-term use in practical applications. The high power output and energy density of our iTEC device and ITEC module with 9-pair legs suggest that our material has potential for use as a self-power source for wearable electronic devices.

Supplementary Table 2 Performance summary of state-of-the-art iTE materials.

Materials	Humidity / %	$ S_i $ / mV K ⁻¹	σ_i / S cm ⁻¹	ZT_i	Maximum strain / %	Self-healability	Reference
This work (PEDOT/PAAMPSA/PA)	70	21.9	0.309	12.3	1050	Yes	
	80	25.1	0.562	28.5			
PVA/PEDOT/PAMPS	80	25.0	0.159	7.2	400	Yes	[8]
PVDF-HFP/EMIM:DCA (doped)	85	43.8	0.019	6.1	n/a	n/a	[9]
PVDF-HFP/EMIM:OTf/FS	90	38.3	0.011	2.34	500	Yes	[10]
PVDF-HFP/EMIM:DCA	70	25.4	0.018	1.8	n/a	n/a	[11]
PEDOT/PSS/CuCl₂	80	18.2	0.053	1.54	n/a	n/a	[12]
WPU/EMIM:DCA	90	34.5	0.008	1.3	156	n/a	[13]
PANI/PAAMPSA/PA	90	8.1	0.237	1.04	750	Yes	[14]
PU/EMIM:DCA/BDB	90	31.4	0.007	0.99	300	Yes	[15]
PSSH	70	7.9	0.09	0.025	n/a	n/a	[16]
Polyamide/EMIm:Tf₂N	n/a	1.4	0.009	n/a	n/a	Yes	[17]

Reviewer #2 (Remarks to the Author):

This is a very interesting and well-written manuscript dealing with stretchable and self healable devices based on a polymer complex, including the conducting polymer PEDOT.

On my opinion the manuscript can be a potential candidate for publication in Nature Communications, after addressing the following issues.

1) The title talks about "Giant Ionic Thermoelectric Effect". How can the adjective "Giant" be justified? Maybe a table to compare state of the art results would be helpful.

Response: We greatly appreciate the valuable feedback provided by the reviewer, which prompted us to add Supplementary Table 2 in the Revised Supplementary Materials. This comparison clearly demonstrates that our material has a ZT_i of 12.3 at 70% RH, which is significantly higher than the current state-of-the-art materials at higher RH. For instance, the previous best iTE material exhibited a ZT_i of 7.2 at 80% RH, whereas our PEDOT:PAAMPSA:PA can achieve a ZT_i of 28.5 at 80% RH (refer to Fig. 2b in the manuscript). As the adjective 'Giant' has often been used to express high ionic thermoelectric performance (*Science* **2020**, 368, 1091–1098, *ACS Appl. Mater. Interfaces* **2022**, 14, 17, 19304–19314, and *Adv. Energy Mater.* **2022**, 2200858), we believe that our results justify using the term to describe the outstanding performance of our material.

Supplementary Table 2 Performance summary of state-of-the-art iTE materials.

Materials	Humidity / %	$ S_i $ / mV K ⁻¹	σ_i / S cm ⁻¹	ZT_i	Maximum strain / %	Self-healability	Reference
This work (PEDOT/PAAMPSA/PA)	70	21.9	0.309	12.3	1050	Yes	
	80	25.1	0.562	28.5			
PVA/PEDOT/PAMPS	80	25.0	0.159	7.2	400	Yes	[8]
PVDF-HFP/EMIM:DCA (doped)	85	43.8	0.019	6.1	n/a	n/a	[9]
PVDF-HFP/EMIM:OTf/FS	90	38.3	0.011	2.34	500	Yes	[10]
PVDF-HFP/EMIM:DCA	70	25.4	0.018	1.8	n/a	n/a	[11]
PEDOT/PSS/CuCl₂	80	18.2	0.053	1.54	n/a	n/a	[12]
WPU/EMIM:DCA	90	34.5	0.008	1.3	156	n/a	[13]
PANI/PAAMPSA/PA	90	8.1	0.237	1.04	750	Yes	[14]
PU/EMIM:DCA/BDB	90	31.4	0.007	0.99	300	Yes	[15]
PSSH	70	7.9	0.09	0.025	n/a	n/a	[16]
Polyamide/EMIm:Tf₂N	n/a	1.4	0.009	n/a	n/a	Yes	[17]

2) The authors cite a stretchability of 500% for human skin, which looks a bit exaggerated. I believe a reasonable value would be around 30%.

Response: We appreciate the reviewer's comments. We agreed to the reviewer and revised the manuscript.

Revised manuscript

In addition to mechanical robustness (Young's modulus (Y) of 0.5 to 1 MPa), sufficient stretchability and spontaneous self-repairing ability are required.

Reproducible mechanical stretchability (a strain of >500%) was easily demonstrated in the PEDOT:PAAMPSA:PA free-standing film (Supplementary Movie 1).

3) Figure 2: the caption says that the values in Figure an and C are averaged on 5 and 3 samples respectively. However, no error bars are shown.

Response: We appreciate the reviewer's comments. As the reviewer suggested, we revised Fig. 2a and 2c with error bars.

Revised Fig. 2a and 2c

4) Figure 3b legend are invisible. Suggest to remove them and leave the scale bar only.

Response: We appreciate the reviewer's comments. As the reviewer suggested, we revised Fig. 3b with scale bar.

Revised Fig. 3b

5) The statistics about the mechanical measurements (stress-strain) is unclear. How many

samples have been measured to extract the stretchability values? Please explain the meaning of maximum strain and toughness in Figure 4d and explain how the values were obtained.

Response: We are grateful for the valuable feedback provided by the reviewer. As per their suggestion, we have made revisions to Fig. 4d, incorporating error bars/average/SD. All data is averaged from three samples each. The maximum strain was determined based on the fracture point, while the toughness was calculated by integrating independent stress-strain curves.

Revised Fig. 4c and 4d

Revised caption in Fig. 4d

(d) Average maximum strain and toughness of PAAMPSA:PA and PEDOT:PAAMPSA:PA free-standing films with different PEDOT/PAAMPSA ratios at 60% RH.

Revised manuscript

Fig. 4c shows the stress-strain characteristics of the PEDOT:PAAMPSA:PA free-standing films. The maximum stress in the films decreased for the film with 2.1wt. % PEDOT and consistently increased with the increase in the PEDOT content, demonstrating 450 kPa at 6.2 wt.% PEDOT. The average maximum strain and toughness of the PEDOT:PAAMPSA:PA free-standing films were also enhanced as the PEDOT/PAAMPSA ratio increased (Fig. 4d), owing to the more vital dynamic interaction among the components.

The raw data for the three measurements are shown below.

6) Please give more details about the self healing experiment. How exactly the cuts were performed (e.g. in figure 2b)? How was humidity controlled during the experiment? Would it be possible to define a healing efficiency (from Fig 4g it looks close to 100%)

Response: We sincerely appreciate the reviewer's comments. To conduct our self-healing tests, we used a conventional razor blade (DORCO, PROCUT) to cut thin films on a glass substrate. In contrast, free-standing films were cut using conventional scissors, as demonstrated in Supplementary Movie 3.

To ensure accurate electrical and mechanical measurements, the humidity was carefully controlled within a sealed plastic chamber. We utilized a reliable commercial hygrometer (DAIHAN Scientific, A1.H9213) to monitor the relative humidity levels.

Fig. 4f and Supplementary Fig. 10 show the thermoelectric performance changes by repeated self-healing. Although we are currently not sure how to quantitatively define healing efficiency, we believe the efficiency is qualitatively high even after ~30 times of repeated self-healing.

7) Although it might not be relevant for this work, do the authors know about the electrical conductivity of the materials?

Response: We appreciate the reviewer's comments. As shown in the Figure below, the electronic conductivity of our material is negligible ($\sim 9.5 \times 10^{-4} \text{ S}\cdot\text{cm}^{-1}$) compared to ionic conductivity ($\sim 0.3 \text{ S}\cdot\text{cm}^{-1}$). We believe that the conductivity of our material is dominated by ionic conductivity. We added Supplementary Fig. 4 and revised the manuscript to convey this content to readers.

Supplementary Fig. 4 | ionic conductivity (σ_i) and electronic conductivity of PEDOT:PAAMPSA:PA with different PEDOT/PAAMPSA ratios at 70% RH. The equivalent circuit in the figure was used to determine electronic conductivity from EIS measurement.

Revised manuscript

Meanwhile, the electronic conductivity was negligible regardless of PEDOT content ($\sim 9.5 \times 10^{-4} \text{ S cm}^{-1}$), indicating that the conductivity of our material is dominated by σ_i (Supplementary Fig. 4).

8) I believe that for a such high-level publication, the healing mechanism needs to be better described, although I understand it might not be straightforward. Anyway the authors could discuss the role of the different components of the blend in the self-healing process and try to explain why the healing time seems to strongly depend on the humidity.

Response: We appreciate the valuable feedback provided by the reviewer. The self-healing mechanism of this type of polyelectrolyte complex has been extensively studied in the literature, highlighting the crucial role of reversible hydrogen bonding and Coulombic interactions among its components (PEDOT, PAAMPSA, and PA) as major contributors to the healing process (*Adv. Mater.* **2022**, 34, 2108932). Although it is challenging to precisely distinguish the role of each species, our findings suggest that the PA component enhances these reversible interactions for the following reasons. Firstly, we have previously reported that PANI:PAAMPSA films lacking PA were unable to self-heal under ambient conditions (*Energy Environ. Sci.*, **2020**, 13, 2915-2923). Secondly, as a multivalent crosslinker, PA facilitates the formation of numerous hydrogen bonds and Coulombic interactions, attracting polymer chains from the surrounding environment. Additionally, the intrinsic properties of PA, including its stickiness and viscosity, make it an effective interaction core, as illustrated in Fig. 1a.

The observed correlation between self-healing time and humidity can be attributed to the effects of water absorption. High ambient humidity promotes the incorporation of water molecules into the material, thereby facilitating charge dissociation and the formation of ion transport channels. Our results on carrier concentration and diffusion coefficient (Supplementary Fig. 7) provide evidence supporting this mechanism. As humidity increases, the concentration of reversible interactions and the speed of diffusion also increase, resulting

in a shorter self-healing time. This concept is consistent with previous literature reports (*Adv. Mater.* **2017**, *29*, 1703098).

9) The caption of Fig 4b is a bit misleading since it would mean that the SEM image is taken during the healing process. Please clarify.

Response: We appreciate the insightful comments provided by the reviewer. To perform SEM observation of the self-healing process shown in Fig. 4b, we followed these procedures: 1) cutting the film, 2) conducting SEM measurement (left image), 3) self-healing the film under 70% RH for 30 seconds, 4) conducting SEM measurement (middle image), 5) continuing self-healing under 70% RH until the scar was completely disappeared, and 6) conducting a final SEM measurement (right image). We have updated the caption of Fig. 4b to clarify these procedures.

Revised caption of Fig. 4b

(b) SEM images of each self-healing stage of a PEDOT:PAAMPSA:PA (6.2wt.% PEDOT) thin film on a glass substrate at 70% RH. Scale bar indicates 300 μm .

10) Table 1: please explain how the error is evaluated.

Response: We appreciate the valuable feedback provided by the reviewer. To calculate the errors presented in Table 1, we obtained the standard deviation from five independent measurements for each sample. We have included the raw data below for transparency and reproducibility.

11) The following literature about self healing PEDOT-based materials should be considered.

Y. Li, X. Zhou, B. Sarkar, N. Gagnon-Lafrenais, and F. Cicoira, Recent progresses on Self-healable Conducting Polymers, *Advanced materials*, 2108932, 2022.

X. Zhou, A. Rajeev, A. Subramanian, Y. Li, N. Rossetti, G. Natale, G. A. Lodygensky, and F. Cicoira, Self-healing, stretchable, and highly adhesive hydrogels for epidermal patch electrodes, *Acta Biomaterialia*, 139, 296-306, 2022.

Y. Li, S. Zhang, N. Hamad, K. Kim, L. Liu, M. Lerond, F. Cicoira, Tailoring the Self-Healing Properties of Conducting Polymer Films, *Macromol. Biosci.*, i2000146, 2020.

Y. Li, X. Li, S. Zhang, L. Liu, N. Hamad, S. R. Bobbara, D. Pasini, F. Cicoira, Autonomic Self-Healing of PEDOT:PSS Achieved via Polyethylene Glycol Addition, *Adv. Funct. Mater.*, 30, 2002853, 2020.

S. Zhang, F. Cicoira, Water-Enabled Healing of Conducting Polymer Films, *Adv. Mater.*, 29, 1703098, 2017.

Overall, I suggest publication after major revisions.

Response: We greatly appreciate the reviewer for their insightful comments and for sharing relevant literature on self-healable PEDOT-based materials. We have cited all the relevant references in our manuscript, and we thank the reviewer for their valuable input.

Reviewers' Comments:

Reviewer #1:

Remarks to the Author:

Dear authors,

1. Please add to Supplementary materials how the electronic conductivity was calculated, so it can be informative for less knowledgeable scientists interested in this field.
2. In addition, it will be valuable to add to the supplementary materials representation of an EIS Bode plot, Nyquist plot, and phase, with the fitting circuit that was used to calculate the ionic conductivity, electronic conductivity, and dielectric constant.

Reviewer #2:

Remarks to the Author:

I am satisfied with the revisions. The manuscript can be published without further changes.

Point-by-point Response to Reviewer's Comments

REVIEWERS' COMMENTS

Reviewer #1 (Remarks to the Author):

Dear authors,

1. Please add to Supplementary materials how the electronic conductivity was calculated, so it can be informative for less knowledgeable scientists interested in this field.

Response: We appreciate the valuable feedback provided by the reviewer. Based on the suggestion, we revised the method section in the Supplementary Materials accordingly.

Revised Supplementary Materials

The ionic resistance (R_i) and electronic resistance (R_e) are obtained by fitting the Nyquist plot (**Supplementary Fig. 4**) with the equivalent circuit model (**Supplementary Fig. 6**) using the EIS analyzer software. The electrical conductivity was calculated using the following equation:

$$\sigma = \frac{1}{R} \frac{L}{A} \quad (1)$$

R corresponds to the ionic and electronic resistance, L is the distance between the two gold electrodes, and A is the cross-sectional area of the sample film.

2. In addition, it will be valuable to add to the supplementary materials representation of an EIS Bode plot, Nyquist plot, and phase, with the fitting circuit that was used to calculate the ionic conductivity, electronic conductivity, and dielectric constant.

Response: We appreciate the reviewer's comments. As the reviewer suggested, we added the representative Nyquist plot, Bode plot, and dielectric constant fitting curves in the Supplementary Materials.

Revised Supplementary Fig. 4 | Nyquist plots of PEDOT:PAAMPSA:PA films with different PEDOT/PAAMPSA ratios.

Revised Supplementary Fig. 5 | Bode plots of PEDOT:PAAMPSA:PA films with different PEDOT/PAAMPSA ratios at 70% RH.

Revised Supplementary Fig. 8 | Fitted curves (Red line) of the dielectric constant (Black line) of PEDOT:PAAMPSA films with different PEDOT/PAAMPSA ratios at 80% RH.

Revised manuscript

Meanwhile, the electronic conductivity was negligible regardless of PEDOT content ($\sim 9.5 \times 10^{-4} \text{ S}\cdot\text{cm}^{-1}$), indicating that the conductivity of our material is dominated by σ_i (Supplementary Fig. 4, Supplementary Fig. 5, and Supplementary Fig. 6). Notably, PA negligibly influenced the iTE properties of the films (Supplementary Fig. 7), acting solely as a physical crosslinker. Fig. 2c shows the changes in the net n_i and D_i of the PEDOT:PAAMPSA films with respect to the PEDOT/PAAMPSA ratio. Each net n_i and D_i is determined by fitting the dielectric constant-frequency curve (Supplementary Fig. 8).

Reviewer #2 (Remarks to the Author):

I am satisfied with the revisions. The manuscript can be published without further changes.

Response: We sincerely appreciate the reviewer's kind and positive comment.